# Genome-Wide Characterization of Tomato *FAD* Gene Family and Expression Analysis under Abiotic Stresses

**DOI:** 10.3390/plants12223818

**Published:** 2023-11-10

**Authors:** Rui Xi, Huifang Liu, Yijia Chen, Hongmei Zhuang, Hongwei Han, Hao Wang, Qiang Wang, Ning Li

**Affiliations:** 1Institute of Horticulture Crops, Xinjiang Academy of Agricultural Sciences/Xingjiang Engineering Research Center for Vegetables, Urumqi 830091, China; tsuki_re@163.com (R.X.); mirrorlhf@163.com (H.L.); chenyj0721@163.com (Y.C.); zhuanghongmei86@163.com (H.Z.); hhwei2010@sohu.com (H.H.); wanghao183@163.com (H.W.); 2The State Key Laboratory of Genetic Improvement and Germplasm Innovation of Crop Resistance in Arid Desert Regions (Preparation), Institute of Horticultural Crops, Xinjiang Academy of Agricultural Sciences, Urumqi 830091, China; 3College of Horticulture, Xinjiang Agricultural University, Urumqi 830052, China

**Keywords:** tomato, FAD, bioinformatics analysis, biotic/abiotic stresses, gene expression

## Abstract

The fatty acid desaturase (FAD) gene family plays a crucial regulatory role in the resistance process of plant biomembranes. To understand the role of FADs in tomato growth and development, this study identified and analyzed the tomato *FAD* gene family based on bioinformatics analysis methods. In this study, 26 *SlFADs* were unevenly distributed on 10 chromosomes. Phylogenetic analysis showed that the *SlFAD* gene family was divided into six branches, and the exon–intron composition and conserved motifs of *SlFADs* clustered in the same branch were quite conservative. Several hormone and stress response elements in the *SlFAD* promoter suggest that the expression of *SlFAD* members is subject to complex regulation; the construction of a tomato FAD protein interaction network found that *SlFAD* proteins have apparent synergistic effects with SPA and GPAT proteins. qRT-PCR verification results show that *SlFAD* participates in the expression of tomato root, stem, and leaf tissues; *SlFAD8* is mainly highly expressed in leaves; *SlFAD9* plays a vital role in response to salt stress; and *SlFAB5* regulates all stages of fruit development under the action of exogenous hormones. In summary, this study provides a basis for a systematic understanding of the *SlFAD* gene family. It provides a theoretical basis for in-depth research on the functional characteristics of tomato *SlFAD* genes.

## 1. Introduction

Fatty acid desaturase (FAD) is a critical enzyme in plant lipid metabolism, playing essential roles in plant growth, development, and stress responses [1,2,3,4]. Based on their solubility, plant FADs can be categorized into soluble desaturases (FAB2/SAD) and membrane-bound desaturases (FAD) [5], with no evolutionary relationship between the two of them [6,7,8,9]. FAB2s typically contain two conserved histidine motifs (D/EXXH), while membrane-bound FADs contain three conserved histidine motifs (H(X)34H/H(X)23HH/H/Q(X)2~3HH) [10]. Stearoyl-ACP desaturase (FAB2/SAD) is the only known soluble FAD in the plastid stroma, catalyzing the conversion of stearic acid to oleic acid [11]. Membrane-bound FADs can be further divided into four subfamilies based on their functions, including FAD4, FAD2/FAD6 (ω-6 desaturases), FAD3/FAD7/FAD8 (ω-3 desaturases), and ADS/SLD/DES. Amino acid sequences of FADs within the same subfamily exhibit high conservation [12].

The level of unsaturation in FAD is a key determinant of a plant’s ability to withstand adverse environmental conditions, as it enhances the plant’s resistance to stress by regulating the unsaturation of fatty acids in the body. Studies have shown that *MaFADs* play a crucial role in banana’s response to both high- and low-temperature stress [13]. The overexpression of soybean *GmFAD3A* in rice induced by low temperature improved the cold tolerance and seed germination rate of transgenic plants [14,15]. Functional analysis of the soybean fatty acid desaturase *GmFAD3C-1* gene indicated its close association with linolenic acid content in plants [16]. When the four *AhFAD3* genes from peanuts were expressed in Arabidopsis, each had the effect of increasing the total fatty acid content and relative contribution of ALA in seeds, as well as improving seedling survival under salt stress [17]. *AtFAD2* promotes seed germination and salt tolerance in *Arabidopsis thaliana* [18]. The ectopic overexpression of *AtFAD3* and *AtFAD8* both enhanced drought resistance and osmotic tolerance in transgenic tobacco [19]. *Brassica napus* FAD3 (*BnFAD3*) and Arabidopsis FAD8 (*AtFAD8*) have been shown to increase the linoleic acid/linolenic acid ratio and osmotic stress tolerance [20].

With the continuous development of bioinformatics, a total of 18, 23, 27, 38, and 19 *FAD* gene family members have been identified in Arabidopsis [21], poplar [22], banana [13], eggplant [23], and cotton [24], respectively. Tomato, as a widely cultivated economic crop globally, has suffered from adverse biotic and abiotic stresses that negatively affect its yield. However, its short growth cycle and abundant mutant resources make tomato highly valuable for scientific research. Identifying and analyzing the *FAD* gene family can provide a foundation for exploring high-quality stress-tolerant genes in tomato. In this study, based on the SL5.0 reference genome of tomato Heinz 1706 [10], we conducted a comprehensive genome-wide identification of *FAD* gene family members and studied the differential expression patterns of *FAD* family genes in tomato seedling leaves in response to salt stress. The aim is to unearth candidate salt-tolerant genes in tomato and lay the theoretical foundation for research on the molecular mechanisms of salt tolerance in tomatoes.

## 2. Result

### 2.1. Analysis of Physicochemical Properties of SlFAD Gene Family

In the tomato genome SL5.0, a total of 26 members of the *SlFAD* gene family (*SlFAD19* and *SlFAB7*) encoding *SlFAD* were identified. They were named in order of their chromosomal location as *SlFAD1*~*SlFAD19* and *SlFAB1*~*SlFAB7*. A systematic analysis of their physicochemical properties (Table 1) revealed that the amino acids encoded by *SlFAD* ranged from 166 (*SlFAB6*) to 771 (*SlFAD3*) aa; the molecular weight distribution was between 89,844.76 (*SlFAD3*) and 18,924.74 (*SlFAB6*) kDa; the isoelectric point distribution ranged from 5.72 (*SlFAB1*) to 9.02 (*SlFAD18*); the lipid index distribution was between 69.15 (*SlFAB1*) and 96.73 (*SlFAD19*). Among the 26 family members, only 5 members (*SlFAD4*, *SlFAD9*, *SlFAD10*, *SlFAD11*, *SlFAD19*) were hydrophobic proteins.

The subcellular prediction results of the *SlFAD* gene family showed that the 26 family members were mainly located in the cell membrane, endoplasmic reticulum, chloroplasts, and cytoplasm. Among them, 42.31% of *SlFAD* genes were located in the cell membrane, 30.77% of *SlFAD* genes were located in chloroplasts, and others were located in the endoplasmic reticulum or cytoplasm. These results are also similar to the characteristics of the *FAD* gene family.

### 2.2. Phylogenetic Analysis of the SlFAD Gene Family

A phylogenetic tree was constructed using the maximum likelihood method with protein sequences from Arabidopsis, tomato, and eggplant to study the evolutionary relationship of the *FAD* family. Clustering analysis was performed with 19 and 35 *FAD* genes obtained from *Arabidopsis thaliana* and *Solanum melongena*, respectively, as shown in Figure 1. The phylogenetic tree shows that members of the *FAD* family in the different branches show high bootstrap values (100%), and if the bootstrap value is closer to the initially set number of replicate samples, the higher the confidence in the result, while lower values indicate less confidence in the grouping. The 79 members of the *FAD* gene were divided into two subfamilies: membrane-bound FAD and soluble FAD (FAB). Membrane-bound FAD was divided into five branches (FAD2, FAD3/7/8, FAD6, ADS, and SLD), and the FAB2 subfamily was significantly separated from the membrane-bound FAD subfamily. The phylogenetic tree suggests that FAB2 and FAD may have existed in a common ancestor before the differentiation of monocots and dicots. The branch of FAD2, considered to be an ω-6 desaturase, had the most members, totaling 25, 7 of which were from tomatoes. Next was the FAB subfamily with 19 members, 7 of which were from tomatoes. FAD6 is also considered an ω-6 desaturase, and the FAD6 branch includes one member each from Arabidopsis, eggplant, and tomato. The ω-3 desaturases include FAD3, FAD7, and FAD8, with a total of nine members, three of which are from tomatoes. The SLD branch catalyzes the desaturation of sphingolipids at the ∆8 position and includes four members each from tomatoes and eggplants.

### 2.3. SlFAD Gene Location on Chromosomes and Gene Duplication Analysis

Based on the gene location analysis of tomatoes, it can be found that the 26 members of the tomato *FAD* gene family are unevenly distributed on 10 chromosomes (Figure 2), with no *SlFAD* distribution on chromosomes 2 and 9. The number of tomato *FADs* on each chromosome varies significantly, with most genes mainly located at the ends of the chromosomes and multiple tandem repeats present. Chromosome 12 has the most *FAD* genes located, totaling 8; followed by chromosome 6 with 7; chromosomes 1, 3, and 10 each have two; while chromosomes 4, 5, 7, 8, and 11 only have one located. This may indicate that tomatoes have experienced fragment loss during evolution, and the independent evolution and gene duplication of homologous genes have promoted the increase in the number of *SlFAD* members.

Gene duplication is an effective way for organisms to acquire new genes and maintain gene viability. The substitution rate of *FAD* homologous gene pairs was calculated using KaKs Calculator 2.0 (Appendix A). The results showed that there were seven homologous gene pairs in the tomato *FAD* gene family. Among them, *SlFAD1*/*SlFAD19* and *SlFAD4*/*SlFAD10* homologous gene pairs came from chromosome segment duplications, the rest of the duplication pairs originated from tandem duplications, and the gene pairs of the tandem duplications mainly came from multiple duplication pairs on chromosome 12. The Ka/Ks of all seven homologous gene pairs of the tomato *FAD* gene family were <1, indicating that these homologous genes were subjected to environmental solid stresses and that gene evolution and protein function were stabilized.

### 2.4. Analysis of SlFAD Conserved Motifs and Gene Structure

An analysis of the conserved structural domains of the tomato *FAD* family resulted in a total of 10 motifs (Figure 3A), with a high degree of conservation between the same subfamily and the same branch. The evaluation analysis of the motifs indicates that the reverse motif types, quantities, and distributions of FAD proteins belonging to the same subfamily are more similar. Motifs 1, 2, 3, 4, and 7 are present in all FAD2 branches; FAD3/7/8 branches all contain motifs 4 and 7, while motifs 8, 9, and 10 are only found in the SLD branch. This suggests that genes in different branches have a close phylogenetic relationship. There are significant differences in conserved motifs between membrane-bound FAD and FAB. For example, all members of the FAB subfamily contain motifs 5 and 6, except for *SlFAB6*, and these two motifs only appear in the FAB subfamily. This may indicate that members of the *FAD* family were involved in two directions during evolution; one is the FAB subfamily, and the other is the membrane-bound FAD subfamily.

The distribution of introns/exons in each *SlFAD* gene family was further analyzed (Figure 3B). The results showed that *SlFAD* genes clustered into the same branch determined a relatively similar gene structure pattern. The number of introns in the 26 *SlFAD* gene family members ranged from 1 to 9, and the number of exons ranged from 1 to 10. The family members of the FAD3/7/8 branch had the most exons, all above seven. Generally, SLD members only contain one exon and no introns, which is consistent with previous studies, with the least number of exons in the SLD branch, all having one–two. In addition, *SlFAD14*, *SLFAD2*, and *SLFAD13* all have only one exon. Similar *FAD* gene structures were found in the same branch, which also indicates that exon–intron distribution supports the phylogenetic classification of *FAD*. We also examined the intron phases associated with codons. Intron phases were very conserved among intragroup members, whereas intron arrangements and intron phases were significantly different between groups (Figure 3B). This may provide support for the results of phylogenetic and genomic duplication analyses. The amino acid composition of the his-box is highly conserved among members belonging to the same subfamily (Figure 4). A conserved histidine box was found in the FAB subfamily members. Except for *SlFAB6*, all subfamily members contain EENRH and DEKRH, while all members of the *FAD* subfamily contain two conserved histidine boxes, H(XX)H and H(XX)HH.

### 2.5. Analysis of Collinearity Relationships of SlFAD Gene Family

To further investigate the evolutionary process among different species, we characterized duplicate gene pairs among *SlFAD*, *AtFAD*, and *SmFAD* members (Figure 5). The colinearity results showed that all 7 *FADs* in tomato were colinear with the genomes of Arabidopsis and eggplant. Two of them belonged to soluble *FAB* gene pairs, and the rest were membrane-bound *FAD* gene pairs. 10 colinear gene pairs existed for 8 genes from tomato and Arabidopsis (*SlFAD4*/*AtSLD1.1*, *SlFAD4*/*AtSLD1.2*, *SlFAD5*/*AtFAD3*, *SlFAD7*/*AtFAD7*, *SlFAD7*/*AtFAD8*, *SlFAD8*/*AtFAD6*, *SlFAD10*/*AtSLD1.1*, *SlFAD10*/*AtSLD1.2*, *SlFAB4*/*AtFAB2.2* and *SlFAB7*/*AtFAB2.6*). There were more colinear gene pairs between eggplant and tomato, with 15 colinear gene pairs among 13 genes between tomato and eggplant (*SlFAD1*/*SmFAD3*, *SlFAD4*/*SmFAD11*, *SlFAB4*/*SmFAB2*, *SlFAB7*/*SmFAB5*, *SlFAD5*/*SmFAD22*, *SlFAD7*/*SmFAD24*, *SlFAD8*/*SmFAD27*, *SlFAD9*/*SmFAD30*, *SlFAD10*/*SmFAD6*, *SlFAD18*/*SmFAD13*, *SlFAD18*/*SmFAD17*, *SlFAD18*/*SmFAD12*, *SlFAD19*/*SmFAD3*, *SlFAD19*/*SmFAD17*, and *SlFAB1*/*SmFAD1*), which may be related to the close affinity of tomato and eggplant. The 7 genes with colinear relationships with Arabidopsis also co-occurred in eggplant, suggesting that these genes may have played an important role in the evolution of the *FAD* family genes.

### 2.6. Analysis of Cis-Acting Elements in SlFAD Promoters

Further analysis was conducted on the *cis*-acting elements controlling the expression of the *SlFAD* gene family members in the upstream transcription initiation sites of *SlFADs* genes, identifying a total of 26 types. Based on function, they can be divided into four major categories: stress response, growth and development, light response, and hormone response (Figure 6). In stress response, the number of drought response factors (MYC, MBS, and as-1) is the highest, totaling 277. The MYC element was detected in all 26 genes and is the most numerous. Among them, there are 13 in the promoter of *SlFAD8*, indicating that the *SlFAD8* gene may be closely related to the drought response of tomatoes. The total number of STRE elements related to stress ranks second, totaling 95. STRE elements were detected in all promoters except for *SlFAD6*. There are a total of 65 response elements related to anaerobic induction, and this type of element is present in the promoters of all genes except for *SlFAD6* and *SlFAD12*. In addition, elements related to low temperature/wound, defense, and stress responses were found in the promoters of 15, 24, and 17 *SlFADs*, respectively. In addition, some elements related to plant growth and development responses were found. The most common are promoter elements related to seed specificity. Others include meristem expression elements (CAT-box), zein metabolism (O2-site), endosperm expression (GCN4_motif), and circadian-related *cis*-elements. Light response elements were identified in all *SlFADs* promoters. Among them, BOX4 has the most (155), followed by G-box (110). Except for *SlFAD6*, *SlFAD8*, and *SlFAD11*, this element can be found in all *SlFADs* promoters. In hormone response elements, there are the most methyl jasmonate (MeJA) response elements (132), followed by ABRE response elements (97). Response elements related to MeJA responsiveness (TGACG-motif and CGTCA-motif) were found in the promoters of 20 *SlFADs*. The promoters of 18, 16, and 24 *SlFAD* members contain salicylic acid, auxin, and ethylene response elements, respectively. In addition, all *SlFAD* promoters contain at least two hormone response elements.

### 2.7. Expression Analysis of SlFAD Genes in Different Tissues and Fruit Development Stages

The expression patterns of 24 *SlFAD* genes in different tissues such as mature leaves (ML), young leaves (YL), young buds (YFB), roots (ROOT), hypocotyls (HYPO), meristems (MERI), and cotyledons (COTYL) are shown in Figure 7A. The results show that family members of different subfamilies and different branches all have their specific expression patterns. The eight family members of the FAD2 branch are expressed at relatively high levels in tissues such as mature leaves, hypocotyls, meristems, cotyledons, and roots, among which *SlFAD13* has the highest expression level in roots; *SlFAD7* and *SlFAD8* of the FAD3/7/8 branch have very high expression levels in all tissues, but the expression levels of *SlFAD5* and *SlFAD6* are relatively low; and all genes except *SlFAD4* in the SLD branch are highly expressed. Among the members of the soluble *FAD* subfamily, *SlFAB2*, *SlFAB7*, and *SlFAB4* are highly expressed in all tissues. In contrast, the expression levels of other members are relatively low, among which *SlFAB6* is not expressed in any tissue. Among all 24 family members, *SlFAD1*, *SlFAD8*, and *SlFAD10* have the highest expression levels in various tissues, indicating that these 3 genes may play a vital role in the growth and development of tomatoes.

During fruit development, in the FAD2 branch, except for the *SlFAD1* gene which is highly expressed throughout the development period, the rest are only expressed 10 days after flowering (10DPA1), and are expressed at a low level or not expressed at other times(Figure 7B); the expression levels of family members of the FAD3/7/8 branch significantly increase in the early stage of fruit development (0~10DPA2), and then the gene expression levels significantly decrease; the expression pattern of family members of the SLD branch is basically consistent with that of FAD3/7/8, but the *SlFAD4* gene is only highly expressed at 0DPA, and its expression level is low or basically not expressed at other times. The expression pattern of the soluble *FADs* subfamily is opposite to that of membrane-bound FADs. The expression levels of its members in the late stage of fruit development (20DPA~33DPA) are higher than those in the early stage, suggesting that they may play a role in the accumulation process of linoleic acid in the late stage of fruit maturity. Among all 24 family members, *SlFAD1*, *SlFAD8*, and *SlFAD10* have the highest expression levels at each fruit development stage, suggesting that these three members play an important role in fruit development.

### 2.8. Protein–Protein Interaction Network Analysis of SlFAD Family Members

By constructing the interaction network of tomato FAD proteins, we specifically analyzed the mechanism of action of tomato FAD proteins (Figure 8). We removed some proteins with missing annotations and low degree values. The results show that the highest number of proteins interact with SlFAD8. Among them, SlFAD8 has strong interactions with SlFAD7, SlFAD6, and SlFAB4; SlFAD7 with SlFAB2, SlFAB4, and SlFAB7; and SlFAD1 with SlFAD2. SPA is a protein in which the cell wall surface antigen of Staphylococcus aureus exists. In humans and mammals, the complex formed by SPA binding to IgG also has various biological activities such as anti-phagocytosis and promoting cell division, and plays an important role in the early development of biological membranes [25]. GPAT is a type of membrane-bound enzyme that mainly participates in the storage of lipids and catalyzes the initial steps of glycerolipid biosynthesis, playing an important role in plant growth, development, and stress response [26]. There are also some proteins lacking annotations in the interaction network diagram. They have obvious direct or indirect synergistic effects with SlFAD proteins, but their functions are still unclear.

### 2.9. Real-Time Fluorescent Quantitative Analysis of SlFAD Genes

To study the expression pattern of the tomato *FAD* gene family, we analyzed the expression levels of *SlFAD* genes in different tissues, under abiotic stress, and under three different hormone treatments through qRT-PCR experiments (Figure 9 and Appendix A). The results show that family members of different subfamilies and different branches all have their specific expression patterns in roots, stems, and leaves. In the FAD2 branch, *SlFAD12*~*SlFAD19* are only expressed in roots, and the expression levels are basically consistent. Except for *SlFAD1* and *SlFAD2*, they are basically not expressed or are expressed very little in stems and leaves, which is significantly different from other gene expression patterns. The specific expression of these genes in roots indicates that they participate in root development during tomato growth and development. *SlFAD7* and *SlFAD8* of the FAD3/7/8 branch are highly expressed in leaves, which are 10 times and 54 times that of roots, respectively. In the SLD branch, *SlFAD4* and *SlFAD9* have higher expression levels in stems and leaves, and the expression levels of roots, stems, and leaves in *SlFAD10* and *SlFAD11* are basically consistent. It is worth mentioning that the genes *SlFAB3* and *SlFAB5* in the FAB subfamily are super highly expressed in stems and leaves, which are 260 times, 114 times, 245 times, and 122 times that of roots, respectively, and their expression patterns are similar.

Under the treatment of 200 mmol·L^−1^ NaCl simulated salt stress, many genes also produced positive responses (Figure 10 and Appendix A). The expression levels of *SlFAD6*, *SlFAD9*, and *SlFAD10* were all up-regulated compared with the control group, reaching a peak at 8 h. Among them, the expression level of *SlFAD9* was the most significant compared to the control, which was 33 times that of 0 h. The response levels of *SlFAD1*, *SlFAD2*, and *SlFAD7* showed a trend of first increasing and then decreasing over time. The expression patterns of *SlFAD14*, *SlFAD16*, and *SlFAD17* were just the opposite, with their response levels decreasing first and then increasing over time. The overall expression of members of the soluble *FADs* subfamily was up-regulated, and the time to reach the maximum value varied for different genes. Most genes began to show their regulatory effects after 4 h of treatment. This may indicate that members of this subfamily are closely related to tomato’s response to abiotic stress. The remaining genes were down-regulated to varying degrees.

The response mechanism of tomato *FAD* genes was explored using different concentrations of naphthaleneacetic acid, brassinosteroid, and melatonin (Figure 11). Through analysis, it can be found that all genes are strongly induced by EBR at a concentration of 0.1 mg·L^−1^ during the green mature period, indicating that it plays an important role in the early stage of fruit development (Appendix A). Members in the FAD2 branch are significantly up-regulated under NAA, EBR, and MT hormone treatment as a whole, and they all show obvious regulatory effects during the color-changing period. Among them, the expression level of the *SlFAD17* gene under MT treatment is the highest, which is 18 times that of the control group. Members in the FAD3/7/8 branch are generally up-regulated under different hormone treatments. What is different is that most members show significant differences in the later stage of fruit development, and the response is most intense under NAA and MT treatment at a concentration of 30 mg·L^−1^ and 50 mg·L^−1^ during the red mature period. Most members in the FAB subfamily have a more obvious effect in the early and middle stages of fruit development. Among them, the *SlFAB5* gene has significant differences compared with the control group under three hormone treatments at all times, indicating that the *SlFAB5* gene may regulate all stages of fruit development.

## 3. Discussion

Fatty acid desaturase (FAD) plays an important role in plant growth and development and plant defense [13]. Although there have been reports on the resistance of the FAD2 branch in the membrane-bound FAD subfamily to aphid pests in tomatoes, this study differs in that it identifies in the whole genome family members of tomato soluble desaturase (FAB2/SAD) and membrane-bound desaturase (FAD), and performs expression analysis in different tissues and fruit development stages, as well as expression under different exogenous hormones at four maturity stages of fruit. At the same time, we also found that *SlFAD* family members have a very strong response mechanism under salt stress, which will provide some theoretical support for future comprehensive research on the tomato *FAD* gene family. In this study, 26 members of the tomato *FAD* gene family were identified and analyzed. The number of members is not significantly different from that of Arabidopsis and eggplant *FAD*, but the number is much lower than that of wheat (68) [27], alfalfa (62) [28], and rapeseed (84) [11], indicating that the expansion of the *FAD* gene family has species specificity. This expansion may be related to gene duplication events [29]. Gene amplification plays a crucial role in the generation of family genes, and fragment duplication and tandem duplication are usually related to the driving force of family genes [30]. Most members of the *SlFAD* gene family are located in cell membranes and chloroplasts, with a few located in the endoplasmic reticulum and cytoplasm. This is consistent with the conclusion that fatty acid desaturation occurs through two different pathways in cell membranes and chloroplasts/endoplasmic reticulum, as indicated by previous studies [13].

Phylogenetic analysis shows that the *FAD* family is significantly divided into two subfamilies, including soluble and membrane-bound FADs, which is consistent with previous research [10]. Membrane-bound FADs can be further divided into five branches: FAD2, FAD3/7/8, ADS, FAD6, and SLD, similar to wheat [27]. Tomatoes do not contain the ADS branch, which is the same as rice and bananas, indicating that ADS may have formed after the differentiation of monocots and dicots [13]. Gene structure variation is important for gene evolution [31,32], and its stability is also a prerequisite for maintaining functional effects. Members of *SlFAD* in the same subfamily branch show similar intron/exon structures, and the proteins they encode have similar motif compositions. The number of conserved motifs in members of the same family branch is close (Figure 3), and the distribution of conserved motifs is similar, indicating that they may have similar functions. The number and position of conserved motifs in different branches vary greatly, indicating that members of different branches may play different roles in plant growth and development and stress response processes. Similar results were also found in rice [33], mustard [10], alfalfa [2], and *Brassica napus* [34], indicating that the *FAD* gene family is highly conserved. In addition, five *SlFADs* (*SlFAD2*, *SlFAD10*, *SlFAD11*, *SlFAD14*, and *SlFAD13*) only have one exon, which is similar to the situation where some *FAD* genes in rice lack introns [33]. Existing research shows that the lack of introns may be due to horizontal gene transfer, the duplication of intronless genes, or the lack of reverse transcription transposons [35]. The analysis results of conserved motifs show that all branches of FAD2 in membrane-bound FADs contain Motif1, 2, 3, 4, and 7; branches of FAD3/7/8 all contain motif4 and motif7, while motif8, 9, 10 are only found in the SLD branch. Most members of the soluble FAB subfamily contain motif5 and motif6. The distribution of conserved motifs matches the subfamily distribution of the phylogenetic tree, which once again proves the conservatism in the evolution process of the *SlFAD* gene family.

The study of promoter regions helps to understand the interactions and functions of genes. Transcription factors can bind to the promoters of target genes, which is also crucial for the regulation process of abiotic stress signal pathways [35]. Studies have shown that *FAD* genes play an important role in the regulation mechanism of abiotic stress [22]. This study found that tomato *FAD* family genes have a variety of *cis*-acting elements related to hormones and abiotic stress, suggesting that *FAD* family genes may participate in tomato growth and development through different hormone regulation pathways and are related to the regulation of various abiotic stress responses. In this study, *cis*-acting elements related to hormones (such as jasmonic acid, abscisic acid, auxin, and ethylene) and stress (such as drought, wound, stress, anaerobic, and low temperature) were identified in the promoter regions of *SlFADs*, indicating that the expression of *SlFADs* may respond to several hormones and abiotic stresses. Previous studies have suggested that *FADs* play an important role in enhancing plant tolerance to different environmental stresses, which is also confirmed by this study [36,37,38].

Differences in gene expression play an important role in family genes, and analyzing the expression patterns of *SlFAD* is beneficial to the further exploration of its characteristics and functions. The mRNA levels of *FAD2/2-1* accumulated in peanut seeds and the expression levels of *FAD2-2/6* and *SLD1* in leaves increased [39]. *BjFAD2s* are expressed in all tested tissues, and *CaFAD2s* mainly accumulate in flowers and seeds [40]. Some *LuFAD2s*, *LuFAD3s*, and *LuFAB2s* in flax seeds are highly expressed at all stages of seed development, and *CsFAD* genes are constitutively expressed in cotyledons and leaves [40,41]. In this study, some members of the FAB subfamily showed medium or even low expression in different tissues, which is consistent with the results of Nishiuchi et al. [42]. *SlFAB4* has much higher expression levels in mature leaves and cotyledons than other members, suggesting that this gene may play an important role in plant growth and development during the induction process. Members of the FAD3/7/8 branch, such as *SlFAD8*, show significant accumulation in mature leaves, cotyledons, young leaves, and young buds. In addition, higher transcription levels of *SlFAD10* were found in hypocotyls and meristems, and *SlFAD1* and *FAD13* were found to be significantly expressed in roots. Overall, *SlFAD* gene family members are expressed in all tissues, but they are expressed at higher levels in plant leaves, which is also consistent with previous conclusions [21].

The prediction results of FAD protein interactions in tomatoes indicate that SlFAD proteins are widely involved in various stress-related pathways. They play important roles in stabilizing membrane structure, affecting the composition and accumulation of fats, and promoting plant growth and development. Among them, SPA protein is crucial for photoperiod flowering. It can regulate photoperiod flowering by controlling the stability of the flower inducer CO [43]. SPA protein can regulate gene expression in coordination with COP1, and can inhibit photomorphogenesis by regulating the abundance of downstream TFs in the light signal pathway. In addition, SPA protein may regulate many biological processes and developmental pathways in Arabidopsis in a COP1-dependent and -independent manner [44]. GPAT protein mainly participates in the synthesis of cuticular lipids, membrane lipids, and fats [45]. Cuticular lipids give plants certain mechanical strength, drought resistance, and the ability to resist pathogen invasion. Existing research has found that the overexpression of GPAT can enhance plant salt tolerance and cold tolerance [46].

This study systematically analyzes and predicts the structure and related characteristics of the tomato *FAD* gene family, laying a solid theoretical foundation for exploring the molecular mechanism of *SlFAD* in the process of tomato growth and development.

## 4. Materials and Methods

### 4.1. Experimental Materials

The material used in this study is the cultivated tomato variety M82, and the seeds were provided by the Tomato Breeding Project Team of the Institute of Horticultural Crops, Xinjiang Academy of Agricultural Sciences. Tomato seeds were germinated, and tomato seedlings with consistent growth were selected and cultured in 1/2 Hogland nutrient solution. When the tomato seedlings grew to the 4-leaf stage, they were treated with 200 mmol·L^−1^ NaCl solution for stress treatment. Each treatment was set up with three biological replicates, and leaf samples were taken at 0, 0.5, 2, 4, 6, 12, and 24 h after treatment. After sample collection, they were quickly frozen in liquid nitrogen and stored at −80 °C for later use.

The fruit development variety used for research is the potted seedling of cherry tomato “Jingfan Pink Star No.1”. Different concentrations of hormones were sprayed on the fruit during development, including 2,4-Epibrassinolide (EBR), naphthaleneacetic acid (NAA), melatonin (MT), and a control group (see Table 1). Fruit setting was performed at the fruit expansion stage, that is, experimental fruits for green mature stage, color-changing stage, and red mature stage have been selected. For example, when the fruit grows to the red mature stage, its fruits in the previous three stages have also been sprayed with corresponding concentrations of exogenous NAA, EBR, and MT, that is, cumulative spraying. The sprayed fruits were sampled by area at the expansion stage, green mature stage, color-changing stage, and red mature stage. Each treatment was set up with three biological replicates. After quick freezing in liquid nitrogen, they were stored at -80℃ for later use.

### 4.2. Data Source

The complete genome, protein sequence, and gene annotation gff files of tomatoes were downloaded online (http://solomics.agis.org.cn/tomato/) (accessed on 21 December 2022). The *FAD* family genes of Arabidopsis [21] were searched and downloaded (https://www.arabidopsis.org/) (accessed on 21 December 2022), and the FAD protein sequences of eggplant [23] were downloaded from (https://solgenomics.net/organism/Solanum_melongena/genome/) (accessed on 21 December 2022).

### 4.3. Identification and Physicochemical Property Analysis of SlFAD Gene Family

The protein sequences containing the conserved structural domains of *SlFAD* genes (PF00487) and (PF03405) were searched and downloaded from the Pfam database (http://pfam-legacy.xfam.org/) (accessed on 25 December 2022). The obtained protein sequences were used to build a hidden Markov model using HMMER v3.3.2 software, and the complete protein sequence of the tomato was retrieved. The ncbi-blast-2.12.0+ software was used in conjunction with the FAD protein sequence of Arabidopsis for local blastp search. The candidate protein sequences obtained by the two methods were merged, and the NCBI CD-search database (https://www.ncbi.nlm.nih.gov/search/) (accessed on 25 December 2022) and SMART database (https://smart.embl.de/) (accessed on 25 December 2022) were used for verification, finally obtaining the *SlFAD* family member sequences of tomatoes.

The physicochemical properties of the *SlFAD* gene family, including amino acid length, isoelectric point, and molecular weight, etc., were analyzed through the online website ExPASy (https://web.expasy.org/protparam/) (accessed on 25 December 2022). Subcellular localization prediction analysis was performed using the WoLF PSORT platform (https://www.genscript.com/tools/wolf-psort) (accessed on 25 December 2022).

### 4.4. Construction of Phylogenetic Tree and Collinearity Analysis of SlFAD Family Genes

To analyze the evolutionary relationship between different species, the MUSCLE plugin in MEGA v11.0.10 software was used for multiple sequence alignment, with parameters kept at default values. The output file was used to construct a phylogenetic tree using the maximum likelihood method (bootstrap set to 1000 times), in conjunction with two other species.

The One Step MCScanx program in TBtools v2.019 software was used to analyze the collinearity relationship between *FAD* family genes in different species, and then the Multiple Synteny Plot tool was used for visualization [47].

### 4.5. Analysis of Conserved Motifs, Gene Structure, and Chromosome Location of SlFAD

The MEME v5.10.1 online software was used to analyze the motifs of the *SlFAD* gene family, with the number of motifs searched set to 10. The output was visualized using TBtools software, and the specific conserved histidine sequences were analyzed using GeneDoc v2.7 software. The *SlFAD* family genes were searched in the gene annotation gff3 file to analyze gene structure features and locate chromosome positions. Visualization was conducted using the Visualize Gene Structure and Gene Location Visualize plugins in TBtools v2.019 software.

### 4.6. Analysis of Cis-Acting Elements, Interaction Networks, and Expression of SlFAD

The Gtf/Gff3 Sequences extractor program in TBtools software was used to organize the upstream sequences (2000 bp) of *SlFADs* genes. The organized data were used to predict their *cis*-acting elements using the Plant Care database. The STRING online website was used to predict protein–protein interaction relationships, and *FAD* gene expression data in different tissues and developmental stages were retrieved from the Tomato Functional Genome Database (TFGD).

### 4.7. Expression Analysis of Tomato FAD Family Genes

The method of Tiangen Biochemical Technology (Beijing, China) Co., Ltd.’s RNAprep pure plant total RNA extraction kit (DP432) was followed to extract RNA from tomato leaves, and then ChamQ Universal SYBR qPCR Master Mix (Novozan, Dhaka, Bangladesh) fluorescent quantitative reagent kit was used according to the detailed instructions provided. Specific primers for qRT-PCR analysis were designed using the DNAMAN v6.0.3.99 software online tool. PCR reactions were performed using QuantStudioTM 5 fluorescent quantitative PCR system (Thermo Fisher Scientific, Waltham, MA, USA), with a reaction system of 20 μL, using SlActin as an internal reference gene; gene-specific primers are shown in Appendix A, and the amplification conditions are: 95 °C pre-denaturation for 15 min; 95 °C denaturation for 10 s, 60 °C annealing for 30 s, 40 cycles, and a melting curve program is added. The relative expression levels of genes were calculated by the 2^−∆∆Ct^ method. Each treatment was set up with three biological replicates and three technical replicates, and t-tests were used to analyze significant differences between data.

## 5. Conclusions

In this study, we identified 26 *FAD* family genes from the tomato genome. Phylogenetic analysis shows that *SlFADs* can be divided into two subfamilies, including soluble FADs and membrane-bound FADs. The exon–intron composition and conserved motifs of *SlFAD* are similar within the same branch. In addition, the expression of *SlFAD* genes may be regulated by various factors, including hormones, stress, transcription factors, etc. The members of the tomato *FAD* gene family may be expressed directionally and maintain their main functions during evolution. Some *SlFADs* have potential roles in organ development and adaptive responses to stress. Our results lay a solid theoretical foundation for exploring the molecular mechanism of *SlFAD* in the process of tomato growth and development.

## Figures and Tables

**Figure 1 plants-12-03818-f001:**
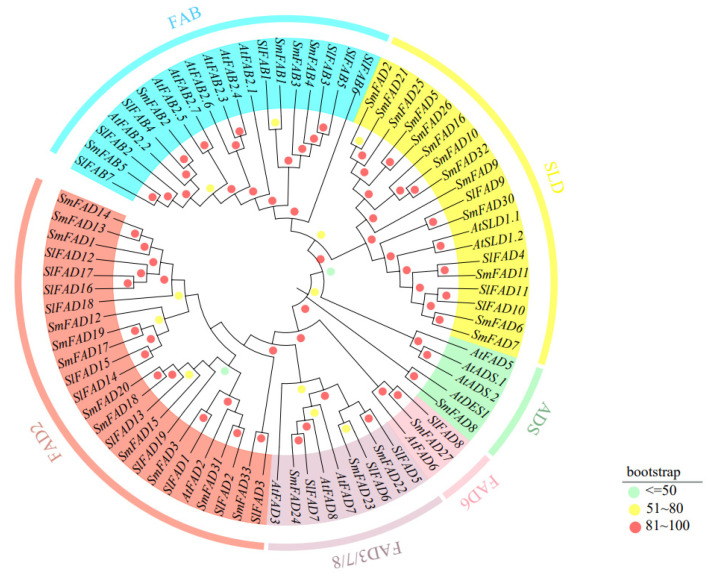
Phylogenetic tree of *FAD* genes. At: Arabidopsis; Sl: tomato; Sm: eggplant. FAD3/7/8: δ-15 desaturase; FAB: stearoyl-ACP desaturase; FAD2/FAD6: δ-12 desaturase; ADS/SLD: anterior end desaturase.

**Figure 2 plants-12-03818-f002:**
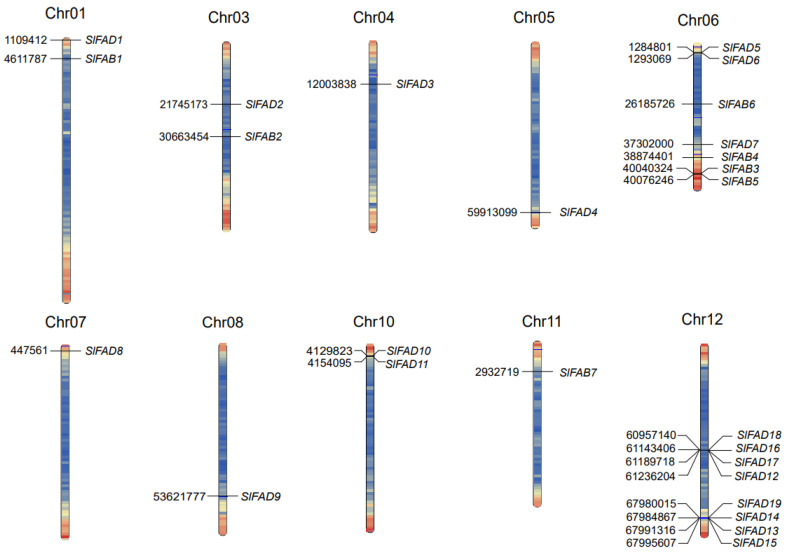
Distribution of *SlFAD* gene family members on chromosomes. Different colors indicate different gene densities; the redder the color, the greater the gene density; the bluer the color, the lower the gene density.

**Figure 3 plants-12-03818-f003:**
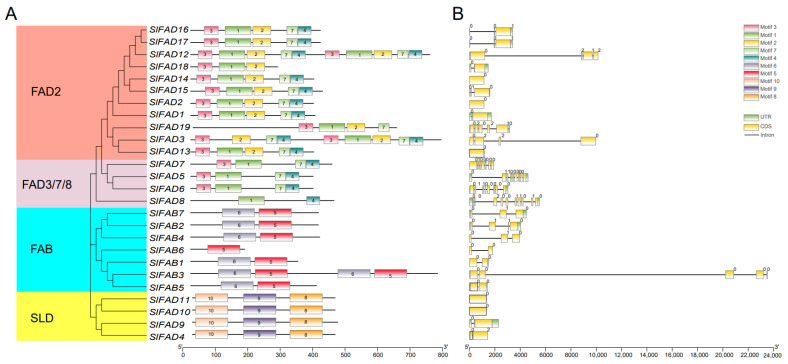
Conserved motifs and structural characterization of *SlFAD*. (**A**) The 10 conserved motifs of the SlFAD protein are represented by colored rectangles, and different branches are marked with different background colors. (**B**) Gene structure of *SlFAD*: CDS indicates coding region; UTR indicates non-coding region; Intron indicates intron.

**Figure 4 plants-12-03818-f004:**
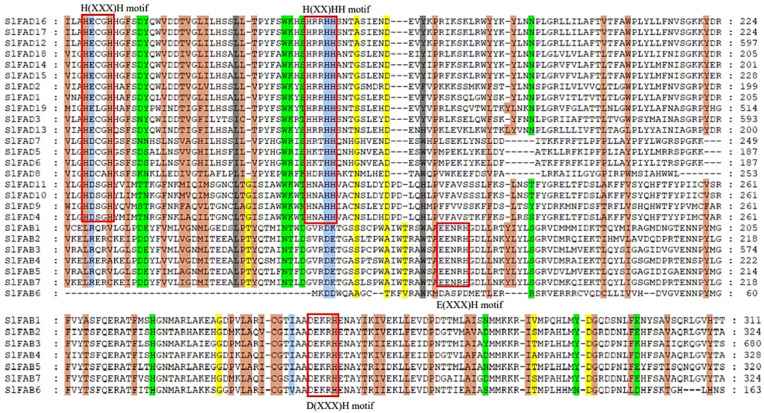
*SlFAD* conserved sequence analysis. Sequence comparison was performed using DNAMAN, and the 3 conserved histidine motifs were marked with red boxes.

**Figure 5 plants-12-03818-f005:**
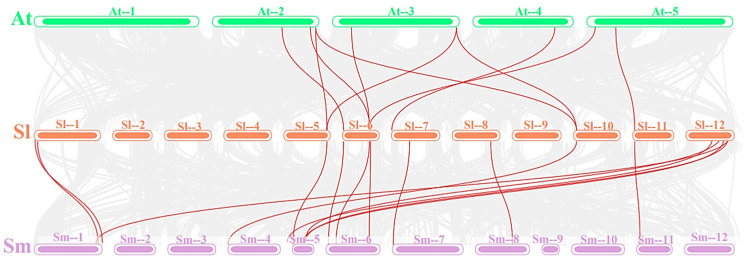
Co-lineage pairs of the tomato *FAD* gene with Arabidopsis and eggplant. Red lines highlight homologous gene pairs of *SlFAD* genes with Arabidopsis, and gray lines indicate genome-wide covariate gene pairs.

**Figure 6 plants-12-03818-f006:**
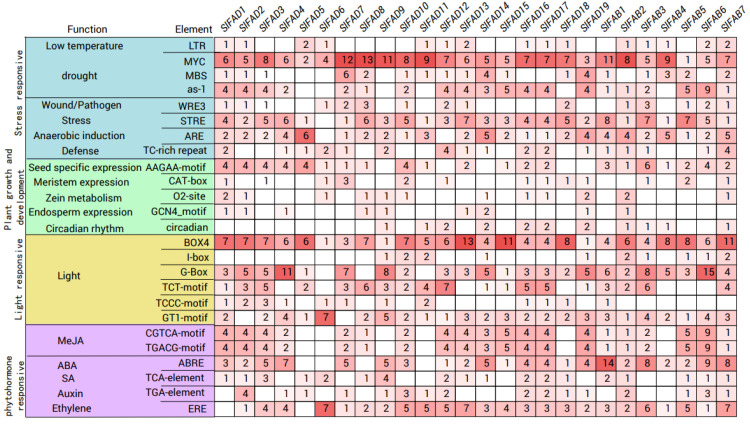
The number of *cis*-acting elements included in the promoters of *SlFADs*. The number of *cis*-acting elements involved in different regulatory pathways varies. A darker red color indicates a higher number of *cis*-acting elements and a lighter red color indicates a lower number of *cis*-acting elements. Different background colors indicate different types of response elements.

**Figure 7 plants-12-03818-f007:**
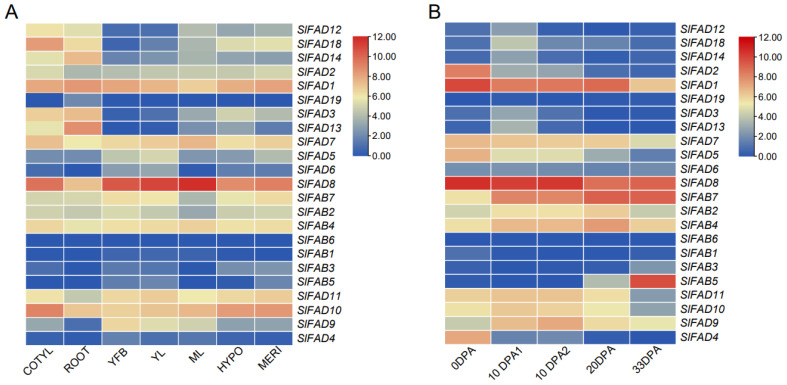
Analysis of expression patterns of *SlFADs* in different tissues and during fruit development. (**A**): expression patterns of *SlFADs* in different tissues; (**B**): expression patterns of *SlFADs* in different fruit development periods; COTYL: cotyledons; ML: mature leaves; YL: young leaves, YFB: young buds; ROOT: roots; HYPO: hypocotyls; MERI: meristematic tissues; DPA: days after anthesis.

**Figure 8 plants-12-03818-f008:**
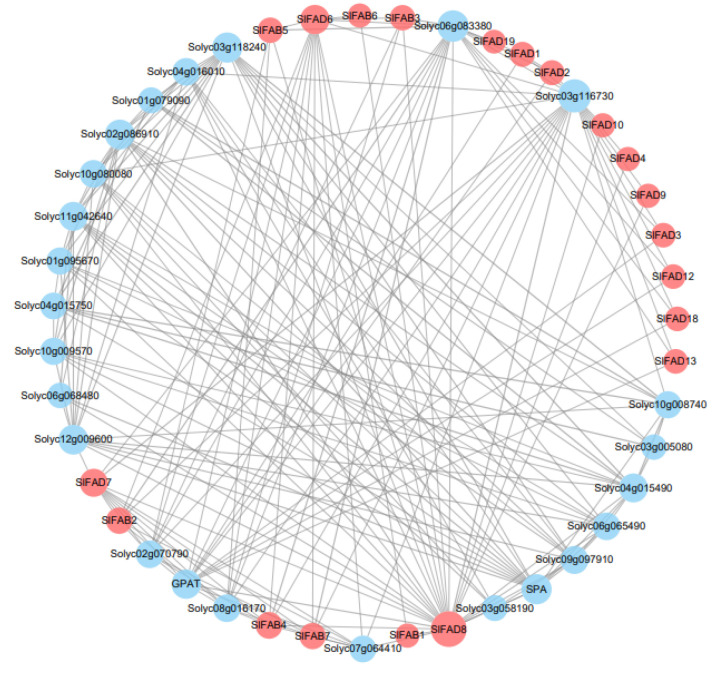
Member of the *SlFAD* gene family protein–protein interaction network. Each node is a protein. Each gray line represents interaction presence, and node size indicates the number of interactions. Red nodes represent SlFAD proteins and blue nodes represent proteins that interact with SlFAD.

**Figure 9 plants-12-03818-f009:**
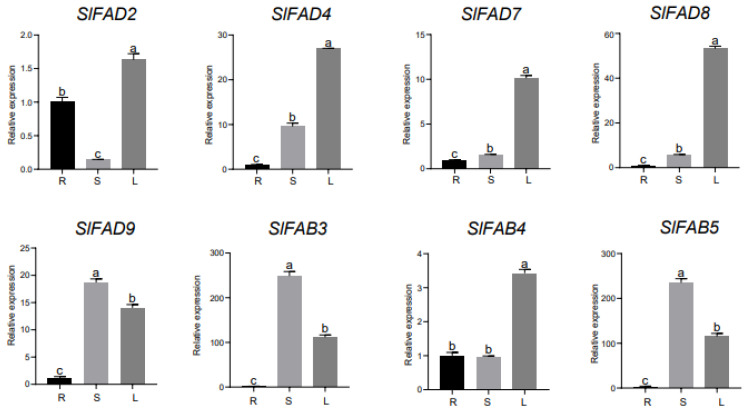
Expression analysis of *SlFAD* gene in roots, stems, and leaves. R: roots; S: stems; L: leaves. a–c: highly significant level of difference (*p* < 0.01). Error bars show the standard deviation of three biological replicates.

**Figure 10 plants-12-03818-f010:**
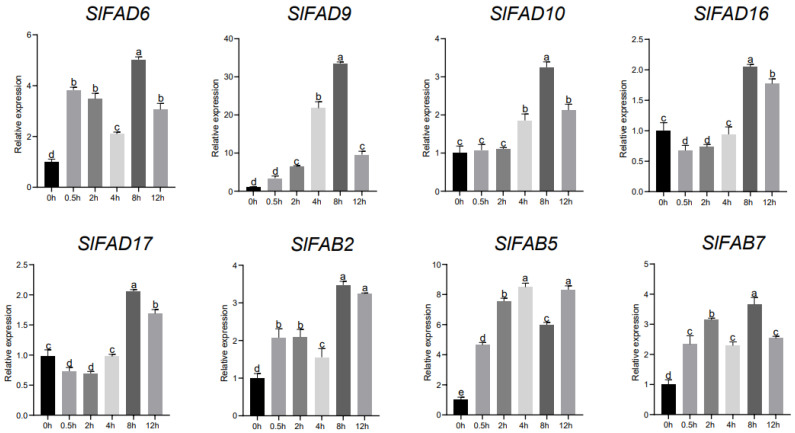
Analysis of *SlFAD* gene expression in leaves under salt stress. a–e: Difference at highly significant level (*p* < 0.01). Error bars show the standard deviation of three biological replicates.

**Figure 11 plants-12-03818-f011:**
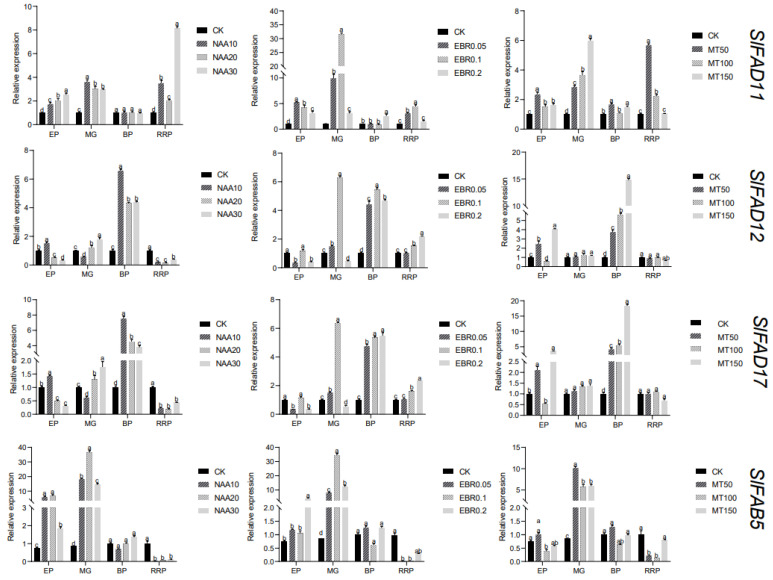
Expression analysis of *SlFAD* gene under hormonal stress during fruit development. a–d: Differences were at highly significant level (*p* < 0.01). Error bars show the standard deviation of the three biological replicates. EP: expansion stage; MG: green ripening stage; BP: color change stage; RRP: red ripening stage.

**Table 1 plants-12-03818-t001:** Physicochemical properties and subcellular localization of *SlFADs*.

Gene ID	Name	Amino Acid Number	Molecular Weight	Isoelectric Point	Fat Index	Hydrophilic/Hydrophobic Proteins	Subcellular Localization
Solyc01T000115.1	*SlFAD1*	383	43,801.58	8.70	90.05	Hydrophilic	cyto
Solyc03T001114.1	*SlFAD2*	378	43,710.22	8.04	87.59	Hydrophilic	plas
Solyc04T000841.1	*SlFAD3*	771	89,844.76	8.56	92.14	Hydrophilic	plas
Solyc05T002114.1	*SlFAD4*	439	50,630.55	8.62	88.18	Hydrophobic	plas
Solyc06T000108.1	*SlFAD5*	377	43,959.96	8.90	90.69	Hydrophilic	plas
Solyc06T000109.1	*SlFAD6*	377	44,085.12	8.45	90.69	Hydrophilic	E.R.
Solyc06T000998.1	*SlFAD7*	435	49,659.73	7.78	83.59	Hydrophilic	chlo
Solyc07T000048.1	*SlFAD8*	441	50,592.87	8.84	89.37	Hydrophilic	E.R.
Solyc08T001312.1	*SlFAD9*	447	51,626.60	8.64	84.79	Hydrophobic	plas
Solyc10T000506.1	*SlFAD10*	439	50,655.63	8.70	89.70	Hydrophobic	plas
Solyc10T000508.1	*SlFAD11*	439	50,577.54	8.68	89.93	Hydrophobic	plas
Solyc12T002062.1	*SlFAD12*	736	86,023.56	8.69	88.22	Hydrophilic	plas
Solyc12T002837.1	*SlFAD13*	379	43,983.43	8.09	95.67	Hydrophilic	chlo
Solyc12T002836.1	*SlFAD14*	379	43,855.47	8.79	89.26	Hydrophilic	E.R.
Solyc12T002838.1	*SlFAD15*	406	47,182.43	8.90	90.52	Hydrophilic	plas
Solyc12T002060.1	*SlFAD16*	399	46,814.10	8.67	94.31	Hydrophilic	plas
Solyc12T002061.1	*SlFAD17*	399	46,814.10	8.67	94.31	Hydrophilic	plas
Solyc12T002056.1	*SlFAD18*	268	31,033.89	9.02	96.04	Hydrophilic	cyto
Solyc12T002835.1	*SlFAD19*	627	70,440.21	8.99	96.73	Hydrophobic	cyto
Solyc01T000424.1	*SlFAB1*	330	37,729.03	5.72	69.15	Hydrophilic	chlo
Solyc03T001283.1	*SlFAB2*	393	44,830.14	6.14	77.43	Hydrophilic	chlo
Solyc06T001235.1	*SlFAB3*	760	87,195.34	6.13	85.33	Hydrophilic	chlo
Solyc06T001110.1	*SlFAB4*	397	45,237.55	6.04	79.65	Hydrophilic	chlo
Solyc06T001237.1	*SlFAB5*	387	44,425.98	7.03	85.17	Hydrophilic	chlo
Solyc06T000610.1	*SlFAB6*	166	18,924.74	8.31	74.04	Hydrophilic	cyto
Solyc11T000349.1	*SlFAB7*	393	44,534.04	6.24	83.38	Hydrophilic	chlo

chlo: chloroplast; cyto: cytoplasm; E.R.: endoplasmic reticulum; plas: cell membrane.

## Data Availability

The supporting data involved in this article are all original, and can be provided upon request.

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
