# Peer review of "Genome-Wide Characterization of Tomato FAD Gene Family and Expression Analysis under Abiotic Stresses"

_plants, 2023, doi:10.3390/plants12223818_

Round 1

Reviewer 1 Report

Comments and Suggestions for Authors

Authors are suggested to Italicize botanical name throughout the manuscript.

Follow standard and consistent pattern of italicizing gene name in the manuscript. e.g., check line no. 57.

Could not find any supplemental file with manuscript while at many places in the text it has been cited. Please check.

Keep Table 2 ( primer list) as supplementary

file.

Comments on the Quality of English Language

Minor editing of English language required

Reviewer 2 Report

Comments and Suggestions for Authors

This manuscript constitutes a nice piece of comprehensive analysis of gene familes. The target gene (FAD) representatives has been discovered in tomato plant genome and thoroughly characterized by both bioinformatics and wet approaches. The inferred characteristics were complemented by real-time analysis of stress-induced changes in these genes. The manuscript is wery well written and the illustrations are informative and quality-made. The only question I have is: what were the biological replications? Single individual plants?  

Reviewer 3 Report

Comments and Suggestions for Authors

This manuscript titled “Genome-Wide Characterization of Tomato FAD Gene Family and Expression Analysis under Abiotic Stresses” presents essential new data about FAD gene family. The manuscript indicates that the FAD gene is ambiguous regarding tomato. There are several shortcomings for that should be resolve.

Some minor remarks:

Lines 111 or, Figure 1: Could you please explain more about the legend of three bootstrap in result or discussion part? Like: A high bootstrap value suggests strong support for the grouping at that node, while a lower value indicates less confidence in the grouping.

Lines 120: Would you kindly analyze the duplication of genes? such as the number of tandem and segmental duplication events in genes. Then, if you discovered any, create a supplementary table.

Lines 141: Figure 3A: Make a representable number inside the motif box. Figure 3B:  Please add intron phases in the gene structure and discuss in the result section.

Lines 169, 170, 297: On the basis of collinear gene please make some correlation between the published paper and FAD gene family in tomato. Please create a hypothetical representative pathway as a diagram based on the facts on hormonal expression and cis-acting element. For a better representation, you might include your hypothetical pathway together with some published pathways.

Lines 180, and so on: Please write “cis-acting elements” instead of “cis-acting element”

Lines 353: “dicots [13].Gene structure” should be “dicots [13]. Gene structure”

Lines 362, and so on: Put "Brassica napus" in italics for the scientific name, or just use "Rapeseed."

Lines 428: Please change the format: “200 mmol·L−1” should be “200 mmol·L−1” or “200 mmol/L”

Line 445, and so on: It’s better to use a weblink, references, and accessed date where it is applicable, like: The complete genome, protein sequence, and gene annotation gff files of tomatoes were downloaded online (http://solomics.agis.org.cn/tomato/) (accessed on 17 September 2023).
